# Investigation of Parameter Sensitivity and the Physical Mechanism for the Formation of a Core-Skin-Core (CSC) Structure in Two-Stage Co-Injection Molding

**DOI:** 10.3390/polym14214747

**Published:** 2022-11-05

**Authors:** Chao-Tsai Huang, You-Ti Rao, Kuan-Yu Ko, Chih-Chung Hsu, You-Sheng Zhou, Chia-Hsiang Hsu, Rong-Yue Chang, Shi-Chang Tseng, Likey Chen

**Affiliations:** 1Department of Chemical and Materials Engineering, Tamkang University, New Taipei City 251301, Taiwan; 2Department of R&D, CoreTech System (Moldex3D) Co., Ltd., Chupei, Hsinchu 302082, Taiwan; 3Department of Mechanical Engineering, National Yunlin University of Science and Technology, Yunlin 64002, Taiwan; 4Material and Chemical Research Laboratories, Industrial Technology Research Institute (ITRI), Hsinchu 310401, Taiwan

**Keywords:** co-injection molding, Core-Skin-Core (CSC) structure, skin break through, lightweight technology, fiber-reinforced plastics (FRP)

## Abstract

One of the main challenges in co-injection molding is how to predict the skin to core morphology accurately and then manage it properly, especially after skin material has been broken through. In this study, the formation of the Core-Skin-Core (CSC) structure and its physical mechanism in a two-stage co-injection molding has been studied based on the ASTM D638 TYPE V system by using both numerical simulation and experimental observation. Results showed that when the skin to core ratio is selected properly (say 30/70), the CSC structure can be observed clearly at central location for 30SFPP/30SFPP system. When the skin to core ratio and operation conditions are fixed, regardless of material arrangement (including 30SFPP/30SFPP; PP/PP; 30SFPP/PP; and PP/30SFPP systems), the morphologies of the CSC structures are very close for all systems. This CSC structure can be further validated by using μ-CT scan and image analysis technologies perfectly. Furthermore, the influences of various operation parameters on the CSC structure variation have been investigated. Results exhibited that the CSC structure does not change significantly irrespective of the flow rate changing, melt temperature varying, or even mold temperature being modified. Moreover, the mechanism to generate the CSC structure can be derived using the melt front movement of the numerical simulation. It is worth noting that after the skin material was broken through, the core material travelled ahead with fountain flow to occupy the flow front. In the same period, the proper amount of skin material with certain inertia of enough kinetic energy will keep going to penetrate the new coming core material to travel until the end of filling. It ends up with this special CSC structure.

## 1. Introduction

Plastics have been widely applied into our daily life because of their great functionalities and flexibility. However, thousands of plastics become solid waste after daily use. It is estimated that more than 150 million tons are sent to the landfill every year [1,2,3]. This plastic waste problem has had a huge impact on our environment. To relieve this kind of huge impact, properly recycling of plastic waste could be helpful. However, due to recycling systems and many issues, the recycling rate is still less than 30% around the world. In fact, recycling of post-consumer plastic waste comes with a lot of challenges, including that the sources are unknown and the recycled wastes are very difficult to separate completely. Moreover, to reduce the air pollution problem, lightweight technology has become one of the most efficient methods. Specifically, one of the major lightweight technologies is utilizing fiber-reinforced plastics (FRP) material into industry for automotive or aerospace products [4,5,6,7], and for modern electric cars [8]. In addition, FRP also can be considered an eco-friendly material by using biocompatible polymer, but in reality, the amount is still small [9]. Hence, the global demand for carbon fibers (CF) and glass fibers (GF) to make FRP is increasing rapidly [10,11,12]. Unfortunately, due to the presence of fibers, the recycling of FRP becomes much more complicated. It will allow our environment to become worse and worse [12,13].

Moreover, co-injection molding could be a good solution to handle recycled general plastic waste or recycled FRP following direct structural composite recycling. Co-injection molding has been proposed and enhanced for several decades [14,15]. The greatest advantages of co-injection are combining materials to reduce cost, reuse material, and enhance production efficiency. However, it is not easy to obtain good quality co-injected products all the time. One of the major challenges is that the morphology of the core material (called the skin–core interface) in the co-injection molding is very difficult to predict accurately and be controlled properly. This morphology affects the aesthetics and properties of the co-injected products significantly. To realize how the morphology of the core material changes, the advancement of the core material should be monitored accurately. In fact, many parameters influence the advancement of core material, including the skin to core material arrangement, mold and part designs, operation conditions, and so on. Seldén [16] studied the key parameters including skin and core temperature and core content to affect the co-injection process. He found that the most significant parameters that affect the skin to core distribution are injection velocity, core temperature, and core content. Furthermore, the relations between internal material distributions, process condition, and material property were discussed in many previous studies [17,18,19,20]. The skin to core ratio of materials determines the breakthrough location. Material viscosity and filling rate affect the uniformity of the core material distribution. Although the operation parameters are not easily managed, co-injection molding can be used to elevate the mechanical properties of injected parts, compared to conventional or self-reinforced injection products [21]. It also can be utilized to enhance the mechanical property of the microcellular injected parts [22], or eco-friendly or new green composite fabrication by using pure biocompatible polymer or polymer blends [23,24].

Moreover, because the complex dynamic behavior of the co-injection process is not easily observed experimentally, several researchers have tried to develop numerical algorithms to discover the evolution of the interface and the internal mechanism [25,26,27,28]. Ilinca and Hetu [25] proposed the simulation code based on finite element method to predict the flow behavior in gas-assisted and co-injection molding. Liu et al. [26] developed a 3D numerical scheme based on generalized Navier–Stokes equations to describe the filling behavior of a co-injection molding process. Sun et al. [27] adopted numerical simulation on a spiral mold to verify the experimental concept of breaking through at [17] successfully. Kim and Isayev [28] proposed a numerical scheme based on a hybrid finite element with finite difference and control volume method to estimate the flow-induced residual stresses and birefringence in sequential co-injection molding. Recently, He et al. [29] utilized CAE simulation and experiments to study the interface of skin to core in a co-injection system. They tried to define the dynamic viscosity ratio of skin and core based on the simulated shear rate results. Then, they claimed that through the adjustment of the dynamic viscosity ratio, the skin–core interface of a co-injection molding can be optimized.

Furthermore, since the geometrical structure of real products is much more complicated, it makes the progress of co-injection processing much more difficult. To consider this situation, Yang and Yokoi [30] proposed a co-injection with multi-cavity molding system with a fork structure. They found the core flow pattern in the fork structure is strongly affected by the injection flow rate. It is also affected by material property. Later, Huang [31] and Huang et al. [32] verified that in multi-cavity co-injection systems, the skin to core material ratio is still the main factor that dominates the breakthrough phenomena, whereas the injection flow rate can be used to adjust the core penetration uniformity. Several key factors have even been discussed to influence the core material advancement on the co-injection processes from the previous literature, however, the morphology variation of that advancement with the skin to core ratio changes, especially for the breakthrough phenomena, and how to quantify that core material advancement has still not been completed. Moreover, although the skin to core interface variation has been studied by many scholars [14,15,16,17,18,19,20,25,26,27,28], the mechanism of the skin material breakthrough has not been fully understood yet. For example, Watanabe et al. [17] has proposed a more comprehensive description about skin material breakthrough. They have considered different material combinations with various operation conditions in sequential co-injection molding. They tried to apply the concept of four regions for flow to explain the mechanism of the breakthrough of skin material. The breakthrough phenomenon happens as the core material penetrates the skin material and catches up with the skin. The skin material will stop flowing and then the core material will continuously fill the flow domain. Overall, in the previous literature, when the skin material breakthrough happens, the core material will take over the skin material to occupy the flow front region fully. However, in some situations, some portion of skin material will compete with the penetrated core material to generate some special structure in co-injection molding. This kind of special structure (we call it “Core-Skin-Core”) has not been described before to the best of the authors’ knowledge. Furthermore, its mechanism has not been discussed before. If paid more attention, this special structure might provide some function which is similar with the multiple-chamber structures in drug-releasing applications, either with macro-scale or nano-scale systems. It could play an important role in new material or new application development in the future [33,34]. For a better understanding associated with this special structure and its mechanisms, the rest of this paper is organized as follows: The theoretical background is presented in Section 2. Section 3 describes the model and related information. It will discuss the simulation model and experimental equipment separately. Moreover, results and discussion are in Section 4. Finally, the conclusion will be addressed in Section 5.

## 2. Theoretical Background

The numerical simulation for a co-injection molding system was studied using Moldex3D R16^®^ software. Both the skin and core materials are considered compressible, generalized Newtonian fluid. Surface tension at the melt front is neglected. The governing equations for 3D transient non-isothermal motion are:(1)$$\frac{\partial \rho }{\partial t}+\nabla \cdot \rho u=0$$
(2)$$\frac{\partial }{\partial t}\left(\rho u\right)+\nabla \cdot \left(\rho uu+\tau \right)=-\nabla p+\rho g$$
(3)$$\rho C_{P}\left(\frac{\partial T}{\partial t}+u\cdot \nabla T\right)=\nabla \cdot \left(k\nabla T\right)+\eta {\dot{\gamma }}^{2}$$
where ρ is density; ***u*** is velocity vector; *t* is time; τ is total stress tensor; ***u*** is acceleration vector of gravity; *p* is pressure; η is viscosity; *C_p_* is specific heat; *T* is temperature; *k* is thermal conductivity; $\dot{\gamma }$ is shear rate. For the polymer melt, the stress tensor can be expressed as:(4)$$\tau =-\eta \left(\nabla u+\nabla u^{T}\right)$$

The modified-Cross model with Arrhenius temperature dependence is employed to describe the viscosity of polymer melt:(5)$$\eta \left(T,\dot{\gamma }\right)=\frac{\eta_{o}\left(T\right)}{1+{\left(\eta_{o}\dot{\gamma }/\tau^{*}\right)}^{1-n}}$$
with
(6)$$\eta_{o}\left(T\right)=BExp\left(\frac{T_{b}}{T}\right)$$
where *n* is the power law index, $\eta_{o}$ the zero shear viscosity, and $\tau^{*}$ is the parameter that describes the transition region between the zero shear rate and the power law region of the viscosity curve.

A volume fraction function *f_i_* is introduced to specify the evolution of the polymer/air front (*i* = 1) and skin to core front (*i* = 2) interfaces. Here, *f_i_* = 0 is defined as the no-filled region, *f* = 1 as fully-filled region, and finally the interfacial front is located within cells of an *f* value between 0 and 1. The advancement of *f* over time is governed by the following transport equation:(7)$$\frac{\partial f_{i}}{\partial t}+\nabla \cdot \left(uf_{i}\right)=0$$

During the polymer melt filling phase, the velocity and temperature are specified at the mold inlet. Whereas the core material is injected, the flow rate setting is specified at the mold inlet. On the mold wall, the non-slip boundary condition is applied, and a fixed mold wall temperature is assumed.

## 3. Model and Related Information

### 3.1. Simulation Model and Related Information

The geometry model and its associated dimensions of the runner and cavity are shown as in Figure 1. It is based on an ASTM D638 Type V standard specimen with dimensions of 63.5 mm × 9.53 mm × 3.5 mm. The mold base and cooling channel layout are exhibited in Figure 1b. The materials utilized in this study include pure polypropylene (called PP material) and polypropylene with 30% fibers (called 30SFPP material), as shown in Table 1. Furthermore, Figure 2 presents the shear viscosities of materials PP and 30SFPP. There is some significant viscosity difference between those two materials. Specifically, material 30SFPP with fiber content has a higher shear viscosity than that of PP at the same shear rate and temperature. Moreover, the process conditions for basic flow behavior testing are listed in Table 2. Briefly, filling time is 0.3 s; flow rate for both skin and core is 10 cm^3^/s; melt temperature for both skin and core is 230 °C; mold temperature is 35 °C; skin-to-core material switch is at 60% of the total volume. Furthermore, the skin to core ratio setting is changed from 90/10 to 10/90 for the skin to core ratio effect study.

### 3.2. Experimental Model and Related Information

In order to validate the simulation results, real co-injection molding trials were performed. The real system was constructed as shown in Figure 3. The machine was supplied by Ta Ai Machinery Co., Ltd. in Taiwan, model TA-4.0ST-2ST-80T (see Figure 3a). There are two injection barrel and screw systems for preparing skin and core materials individually. In each injection system, the screw diameter is 28 mm; maximum injection amount is 75 g; maximum injection flow rate is 70 cm^3^/s; maximum injection pressure is 1302 kg/cm^2^; and maximum clamping force is 85 tons. Moreover, the real mold system and cavity structure are presented in Figure 3b. The dimension of the real mold is that the cavity is 200 mm × 180 mm × 240 mm, and the cavity is 63.5 mm × 9.53 mm × 3.5 mm. The real process condition settings for basic flow behavior testing, skin to core ratio effect, flow rate effect, and different material arrangement effect are the same as described in Section 3.1.

## 4. Results and Discussion

### 4.1. Skin Break-Through Phenomena Testing

As mentioned earlier, one of the most crucial factors in co-injection molding product development is the determination of the morphology of the core material advancement. Specifically, when the skin breakthrough phenomena happens and the associated further influence on the co-injected product are very difficult to identify. One of the most significant factors to skin breakthrough is the skin to core ratio [16,17,18,19,20,25,26,27,28,29,30,31,32]. To study the skin tocore ratio effect on the location of the skin breakthrough, the 30SFPP/30SFPP material arrangement is first selected. Figure 4a shows the numerical prediction on the advancement of the core material at various skin to core ratio settings at a flow rate of 10 cm^3^/s for both skin and core. When the ratio is changed from 90/10 to 60/40, the higher the core content, and the longer core penetration can be obtained. However, when the skin to core ratio is 50/50, the skin break-through phenomenon happens. That means the core material will go from the internal region to blow through the skin material, and touch the end of boundary of the mold. Furthermore, when the skin tocore ratio changed from 50/50 to 10/90, the skin breakthrough area became enlarged; moreover, the final core penetration location is moved from the end of the cavity to the beginning of the cavity, even to the runner section. Clearly, it shows how the breakthrough location is so sensitive to the skin to core ratio in co-injection molding. Meanwhile, experimental validation was also performed, as shown in Figure 4b. When the skin to core ratio is changed from 90/10 to 60/40, the longer core material penetration is observed. As the skin tocore ratio is changed to 50/50, skin break-through happens. When the core ratio is continuously increased from 50 to 90, the skin breakthrough area is increased and the core material tries to encapsulate the skin material. The trend of the advancement of core material from experimental observation is in good agreement with that of the simulation prediction.

### 4.2. Discovering the Core-Skin-Core (CSC) Structure

Moreover, the internal flow pattern of the core material penetration in the final injected parts can be investigated via the cross-section at the central portion of the neck using numerical simulation as shown in Figure 5a. It is noted that when the skin to core ratio is from 90/10 to 70/30, the core material penetration through the neck portion exhibits a round shape pattern. When the skin to core ratio is continuously changed to 40/60, the flow pattern of the core material becomes square in shape with a skin–core structure. The most interesting issue is that as the core ratio becomes 70%, a special pattern can be observed. Afterwards, the special pattern at the cross section portion can be observed when skin to core ratios are 20/80 to 10/90. To realize what the special pattern is in more details, when the skin to core is 30/70, the cross section of the neck portion (central location) becomes enlarged, as shown in Figure 5b. Obviously, the flow pattern of the cross section of the 30/70 system consists of three-layer structure which is the “outer core material-skin material-inner core material.” This special “Core-Skin-Core” (CSC) structure is not the same as that the regular skin–core structure before breakthrough, or that of the core structure after breakthrough happens. In fact, from the previous literature [14,15,16,17,18,19,20,25,26,27,28,29,30,31,32], this kind of CSC structure has never been mentioned and discussed.

### 4.3. The Influence of Operation Parameters on the CSC Structure Variation

Furthermore, it is curious how the CSC structure is influenced by certain injection factors. In much of the literature [16,17,18,19,20,27,30,31,32], the major factors that affect the skin to core interfacial variation include skin to core material ratio, skin to core viscosity difference, injection speed, melt temperature, and so on. Hence, to evaluate the influence on the CSC structure change, firstly, the different skin to core material arrangement effect is considered, which is related to the viscosity difference effect. Figure 6 show the cross section of the central portion for four different skin to core combinations. In Figure 6a, the cross section of 30SFPP/30SFPP exhibited a “core-skin-core”, three-layer structure as we discussed in Figure 5. Figure 6b presents the cross section of PP/PP, and the result is almost identical with that of 30SFPP/30SFPP in Figure 6a. Moreover, the cross section of 30SFPP/PP and that of PP/30SFPP is listed in Figure 6c,d, respectively. Clearly, their cross sections are quite similar with that of the 30SFPP/30SFPP system.

Meanwhile, the variation of the CSC structure under the influence of different factors can be measured quantitatively. The dimension of the CSC structure at the cross section of the center portion can be defined, as shown in Figure 7. Specifically, in the x direction, Lx_1_ is the length of the inner core material; Lx_2_ is the length of the skin material; and Δx is the length of the outer core material. In addition, in the z direction, Lz_1_ is the length of the inner core material; Lz_2_ is the length of the skin material; and Δz is the length of the outer core material. Based on this definition of dimensions for the skin and core materials, the different material combinations the effect has been utilized through numerical simulation to conduct the variation of CSC structure, including PP/PP, 30SFPP/30SFPP, 30SFPP/PP, and PP/30SFPP. Through the breakthrough testing as described in Section 4.2, the results of different material combination arrangements are quite similar with that displayed for the 30SFPP/30SFPP system in Figure 4 and Figure 5. Furthermore, the quantitative variation of the CSC structure is exhibited in Figure 8a,b. In Figure 8a, with the exception of the 30SFPP/PP system, the dimensions of the inner core materials (Lx_1_) are 2.58 mm and for (Lx_2_) are 2.98 mm for three different material combination arrangements. The dimension of the outer core material (Δx) is 0.099 mm. In addition, from PP/PP to another three arrangements, the variations of the CSC structure in the z direction are almost unchanged. Obviously, what happens to the CSC structure is not sensitive to the different material combination arrangement based on the material selected in this study.

Moreover, the flow rate and temperature effects on the variation of the CSC structure have been considered. In Figure 9a, when the flow rate ratios of skin to core are changed from 10/10 cm^3^/s to 50/50 cm^3^/s, or even to 50/10 cm^3^/s, in the x direction, the dimension of the inner core has a little change, but the skin and outer core dimensions show almost no change. In addition, in the z direction, the dimension of the inner core, skin, and outer core dimensions show no significant changes, as shown in Figure 9b. Here, the injection flow rate of the core material has more driving force to push the skin to core interfacial variation seen in the literature [17,27]. However, it does not drive the CSC structure to make the change significantly.

Furthermore, the melt temperatures of the skin and core have been adopted to evaluate the variation of the CSC structure. In Figure 10, it shows that when the melt temperatures of the skin and core are changed from 210/210 °C to 250/250 °C, or even 210/250 °C, the dimensions of the inner core, the skin, and of the outer core have no significant change in both x and z directions. Similarly, Figure 11 shows that when the mold temperatures of the skin and core are changed from 35 °C to 65 °C, in both x and z directions, the dimensions of the inner core, skin, and outer core do not vary significantly. Overall, the CSC structure is not sensitive to the influence of operation parameters. Once again, in several previous studies [16,17,29] the melt temperature can be adopted to modify the skin to core interface to optimize the skin to core distribution, but it does not provide a significant effect here as well.

### 4.4. Validationof the Occurrence of the CSC Structure

To validate the occurrence of the CSC structure, several experimental studies have been performed. Figure 12 displays the comparison of the cross section at the central portion for the skin to core ratio from 90/10 to 10/90 for PP/PP systems between numerical simulation and experimental observation. Specifically, when the skin to core ratio is 70/30 to 50/50, the cross section presents a round shape. When the skin to core ratio is 40/60, the cross section becomes a rectangular shape. When the ratio is changed into 30/70 and 20/80, the special “Core-Skin-Core” structure can be observed experimentally. Indeed, the simulation prediction and experimental observations are quite consistent. As the simulation predicted from Figure 6, the experimental results for the other three systems (i.e., 30SFPP/30SFPP; 30SFPP/PP; PP/30SFPP) are very close to that in Figure 12.

To further investigate the full image of the CSC structure, the PP/30SFPP with a skin to core ratio of 30/70 system has been selected. Here, to distinguish the boundary between one material to the other using image analysis, the densities of both skin to core materials should not be too close. It is the reason to select the PP/30SFPP system. The injected parts have been examined by using micro-computerized tomography (micro-CT) technology. Micro-CT has been performed using Bruker Skyscan 2211 with 40–190 kV and resolution of five micro-meters supported by MCL Multiscale X-ray CT laboratory, Industrial Technology Research Institute, Taiwan. Then the scanned images were further analyzed through image analysis technology based on Avizo^®^ software [35,36,37]. The associate images of the cross section are selected as in Figure 13. Specifically, Figure 13a shows the internal image at the central plane (XY-plane). It is noted that the original skin material is encapsulated by core material from the central region to the end of the filling region. These phenomena were also observed through the investigation of the images at the XZ-plane along the flow direction from central region (see Figure 13b). Obviously, the original skin material has travelled a long distance at this skin to core arrangement. This evidence is quite consistent with that discovery from using numerical prediction.

### 4.5. Search for the Mechanism of the Occurrence of the CSC Structure

To realize how the CSC structure in co-injection molding happens, the mechanism is investigated through the dynamic behavior of the core material advancement using simulation prediction. Figure 14a exhibits the melt front movement for both skin and core material at a skin to core of 70/30. When the total injected volume is at 70%, the injected material is switched to the core material from the runner system. When the total injected volume is over 80%, the core material penetrates into skin layer around the gate region of the cavity. As more core material is injected, the longer the core is penetrated. However, since the melt front of the skin material is far apart from that of core material, the skin breakthrough phenomenon and the CSC structure never happen. Moreover, in Figure 14b, at a skin to core of 50/50, when the total injected volume is less than 50%, only the skin material is injected. When the total injected volume is at 50%, the injected material is switched to the core material and it penetrates into skin rapidly until achieving 77.8% of the total injected volume. If the total injected volume is larger than 77.8%, the skin breakthrough phenomenon happens as the core material takes over the skin material. At this moment, the melt flow front of the skin material stops at a fixed value. This skin breakthrough phenomenon is quite similar as described in [17,27]. However, when the skin to core ratio is changed to 30/70, the dynamic behavior of the core material advancement is seen, as shown in Figure 14c. In this situation, when the total injected volume is at 30%, the injected material is switched to the core material and it penetrates into skin rapidly until reaching 44.7% of total injected volume. If the total injected volume is larger than 44.7%, the skin breakthrough phenomenon happens. As more core material is injected, the core material will keep advancing in the flow direction, whereas some portion of the original skin material will travel with the core material until 100% of the total volume is filled. Finally, it creates the “Core-Skin-Core” structure (i.e., skin material is encapsulated by core material).

Moreover, it is worth noting that the skin breakthrough situation can be observed in more detail, as shown in Figure 15. After the skin has been broken at 50.2% filled, the subsequent core material will push the melt front movement in a fountain flow style. As some core material touches a solid boundary, some frozen layers (core materials) will be generated. At this moment, the original flowing path of the skin material will be blocked by the fountained core layer. However, since the inertial force of the skin material is still high enough, it will help the skin layer with higher kinetic energy continuously penetrate the fountained core layer and travel in the flow direction, as shown in Figure 15b. Furthermore, even more core material is formed and the fountained core layer is continuously formed, as the moment of the moving skin material can keep it moving. Finally, the special CSC structure is generated, as observed and shown in Figure 13. Through this examination, the melt front of the skin material does not stop at a fixed distance; instead, it drives a long distance. This phenomenon is quite different with that observed in Refs [17,27]. In fact, even the fountained core layer has been formed and it tries to block the advancement of the skin material, and a thin skin layer with higher kinetic energy can penetrate the blocking of core material at the skin to core of the 50/50 system. The reason the skin material stops is because of the physical constraint of the end of the flow domain.

## 5. Conclusions

In this study, a formation of the Core-Skin-Core (CSC) structure has been discovered in a two-stage co-injection molding based on the standard tensile bar (ASTM D638 TYPE V) system. Several key points have been obtained as follows:(1)The CSC structure can be predicted numerically and observed experimentally (using slicing method) from the central location when the skin to core ratio is 30/70 for the 30SFPP/30SFPP system.(2)When the skin to core ratio and operation conditions are fixed, the formation of the CSC structure does not depend on the material arrangements, including 30SFPP/30SFPP, PP/PP, 30SFPP/PP, and PP/30SFPP systems. All systems present close results.(3)When the skin to core ratio is fixed, the influences of flow rate change and of the melt temperature variation on the formation of the CSC structure are not significantly.(4)The full domain of the CSC structure can be validated using micro-CT technology and image analysis methods. It verified the existence of the CSC structure perfectly.(5)The mechanism to generate the CSC structure is that when the skin material is broken through by the core material, and the subsequent core material forms fountained, flowing layer to block the skin material; in the same period, some skin portions with higher inertia force with enough kinetic energy can penetrate the blocking of the fountained core and continuously travel in the flow direction until the end of filling to create the CSC structure.(6)The repeatability of the formation for the CSC structure is very good. It might be possible to extend its application to different fields in the future.

## Figures and Tables

**Figure 1 polymers-14-04747-f001:**
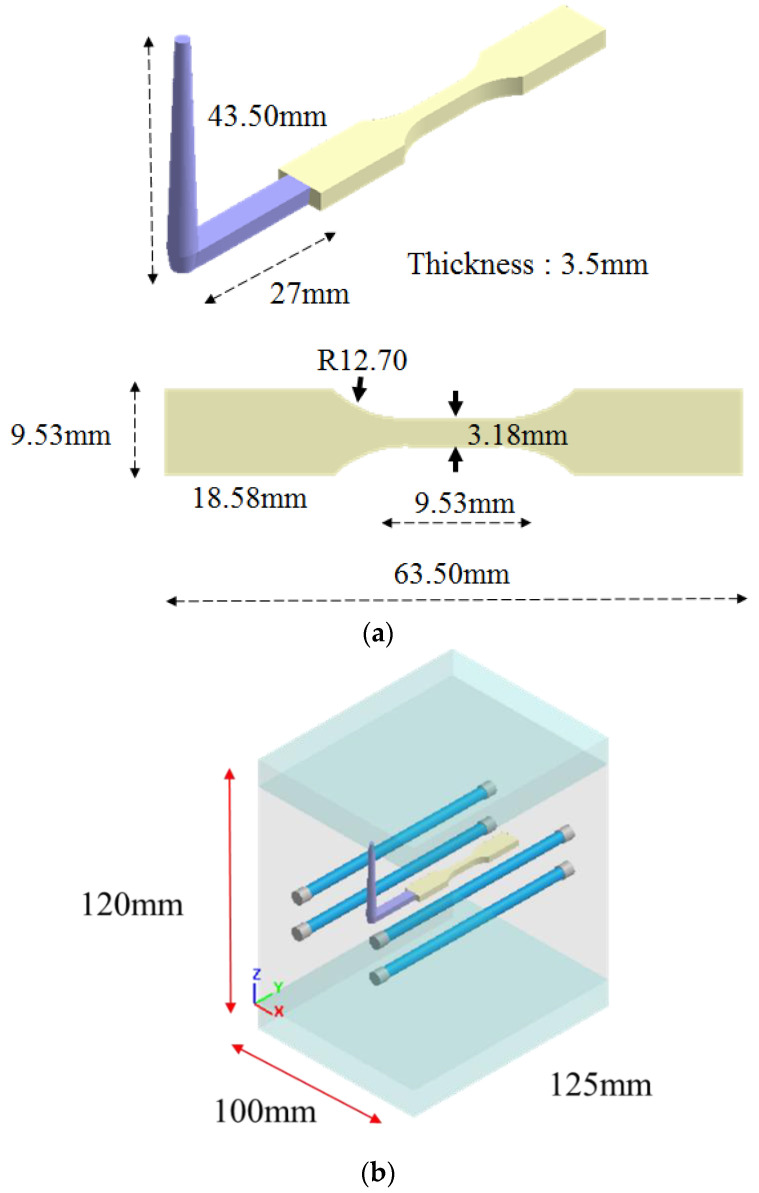
(**a**) Geometrical model and dimensions and (**b**) mold base and cooling channel layout.

**Figure 2 polymers-14-04747-f002:**
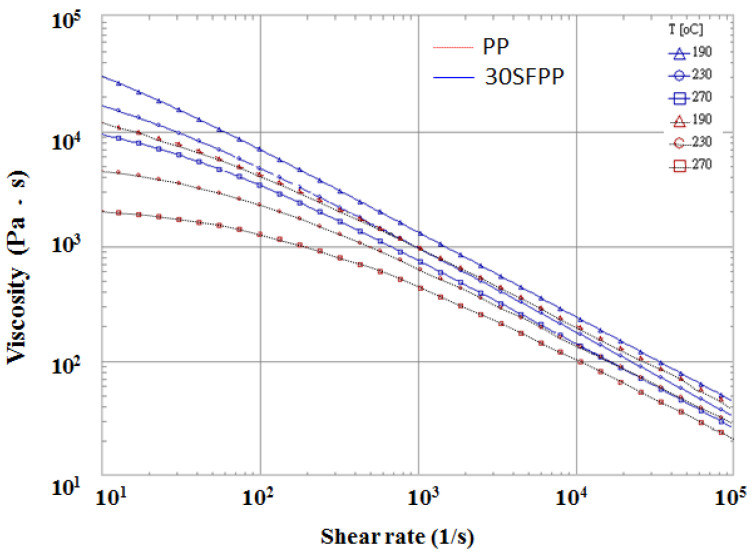
The shear viscosities of material PP and 30SFPP.

**Figure 3 polymers-14-04747-f003:**
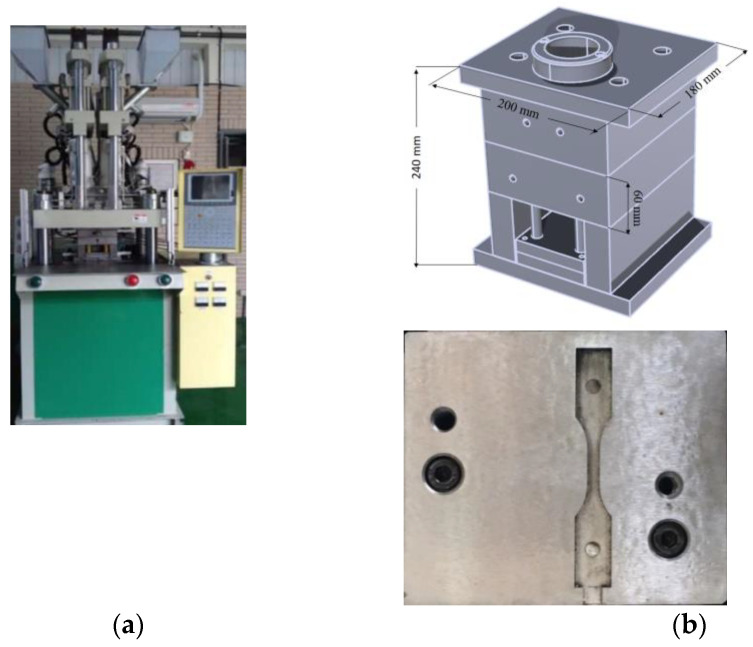
(**a**) Co-injection molding system and (**b**) the mold base and cavity structures.

**Figure 4 polymers-14-04747-f004:**
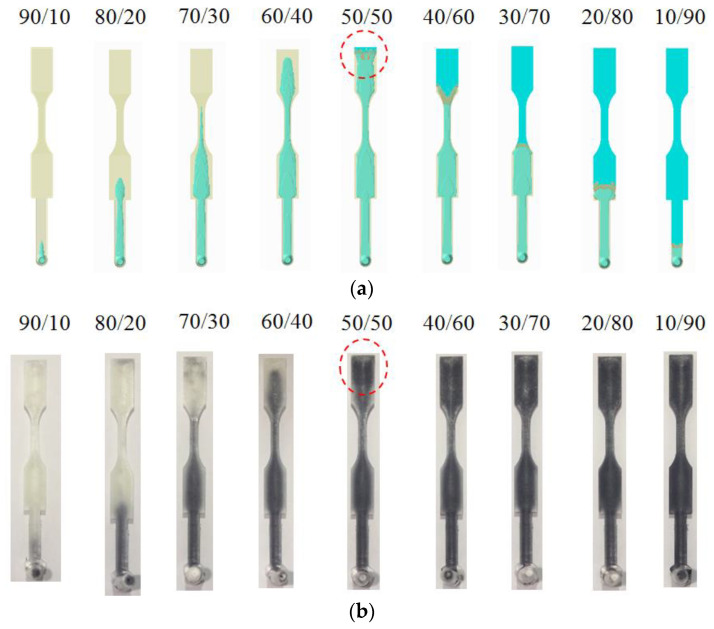
Core material advancement at various skin to core ratio, from 90/10 to 10/90 of 30SFPP/30SFPP arrangement, at 10 cm^3^/s: (**a**) simulation and (**b**) experiment.

**Figure 5 polymers-14-04747-f005:**
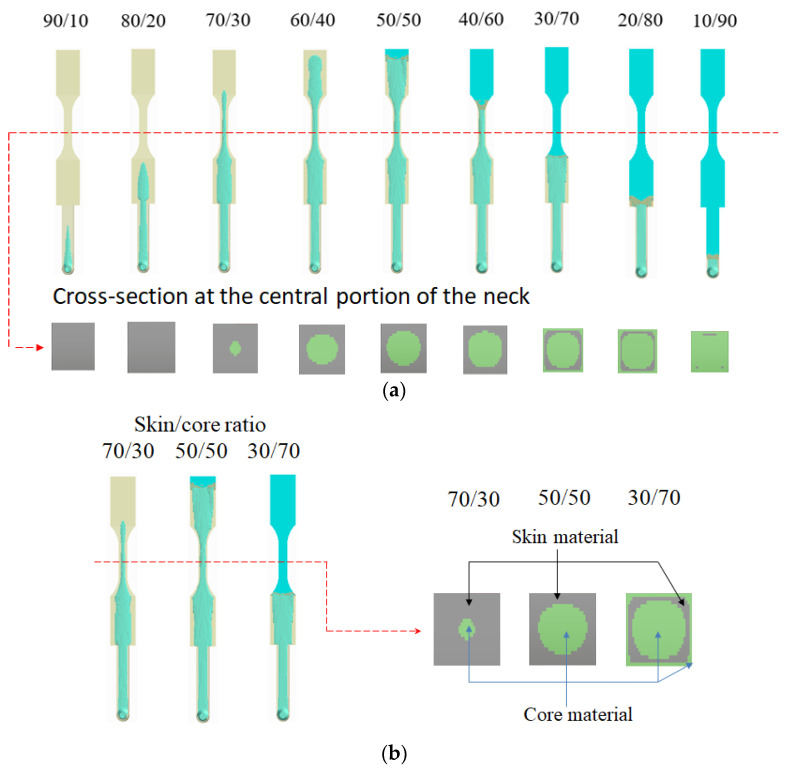
The flow patterns of the core material penetration and their cross section at central portion: (**a**) skin to core ratio from 90/10 to 10/90 and (**b**) the comparison of the flow pattern via the cross section for skin to core ratio of 70/30, 50/50, and 30/70.

**Figure 6 polymers-14-04747-f006:**
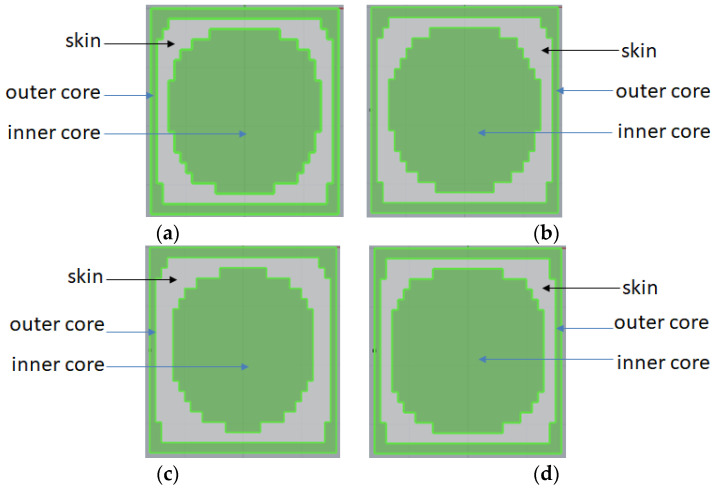
The cross section at center portion for different skin to core material combinations at a skin to core ratio of 30/70: (**a**) 30SFPP/30SFPP; (**b**) PP/PP; (**c**) 30SFPP/PP; and (**d**) PP/30SFPP.

**Figure 7 polymers-14-04747-f007:**
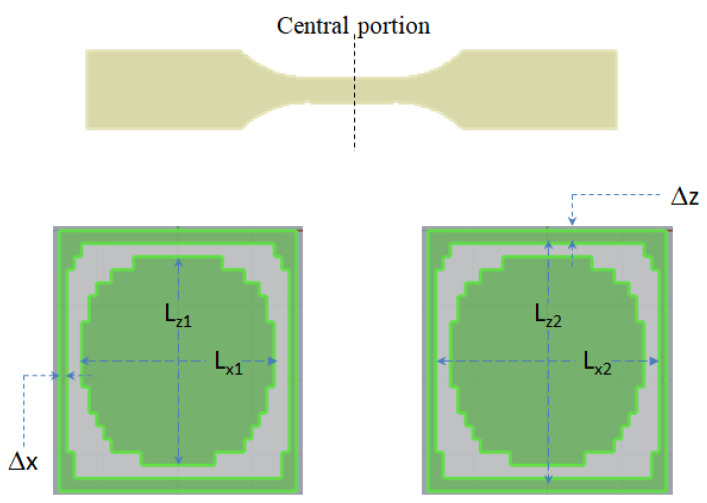
The definition of the dimensions of the core and skin in the cross section of the central portion of the specimen.

**Figure 8 polymers-14-04747-f008:**
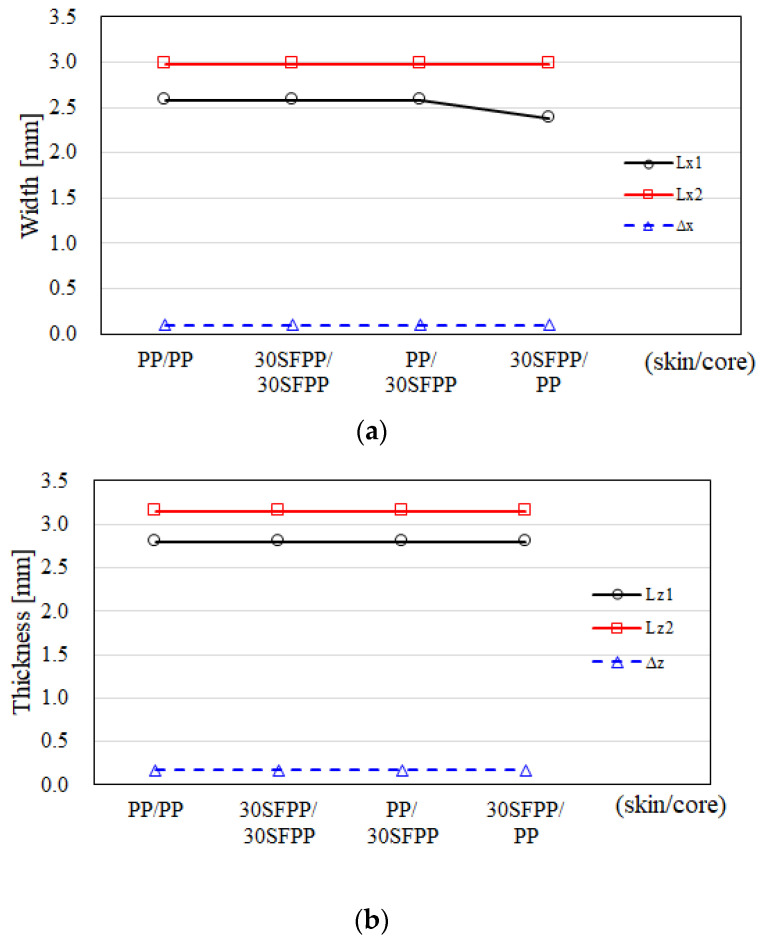
The dimension variation of the CSC structure for different material combinations for skin and core arrangement effects using simulation.

**Figure 9 polymers-14-04747-f009:**
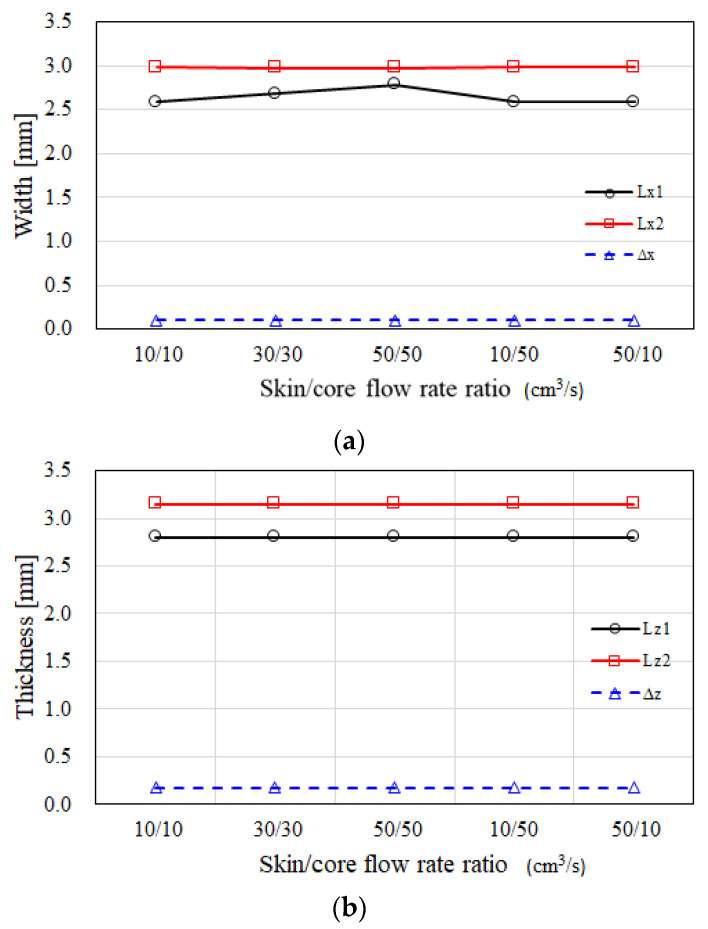
The dimension variation of the CSC structure under the influence of the different skin to core flow rate ratios.

**Figure 10 polymers-14-04747-f010:**
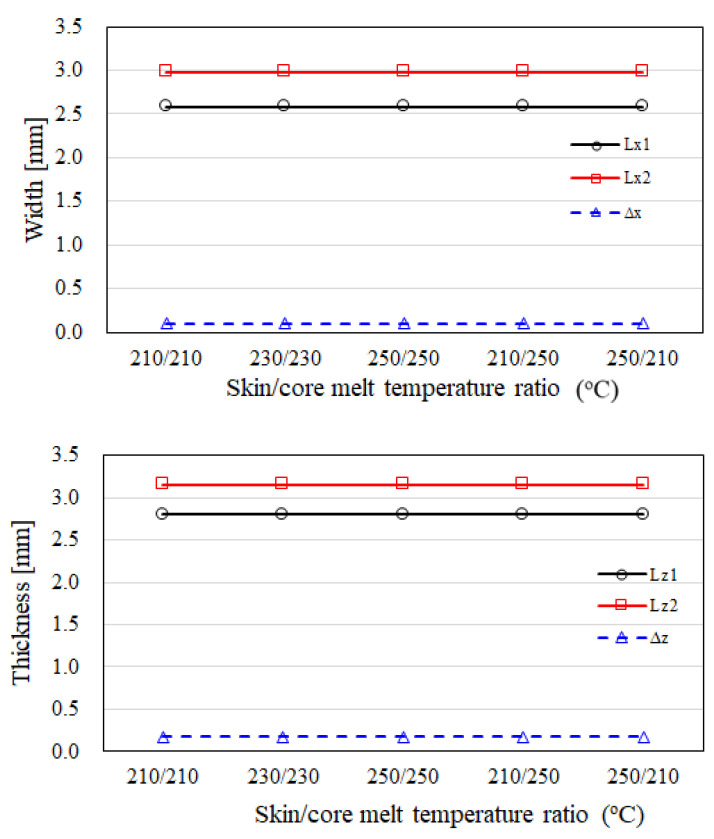
The dimension variation of the CSC structure for different melt temperature effects.

**Figure 11 polymers-14-04747-f011:**
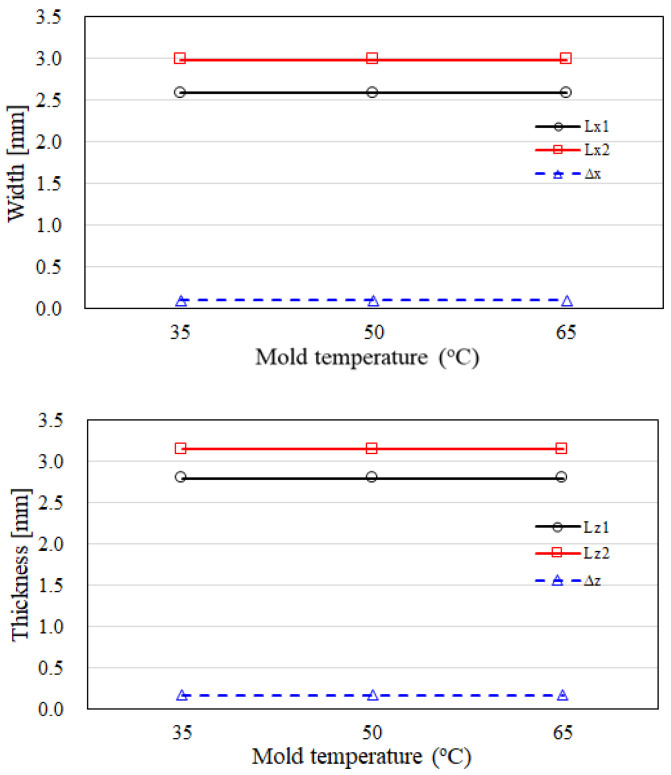
The dimension variation of the CSC structure for melt temperature effects.

**Figure 12 polymers-14-04747-f012:**
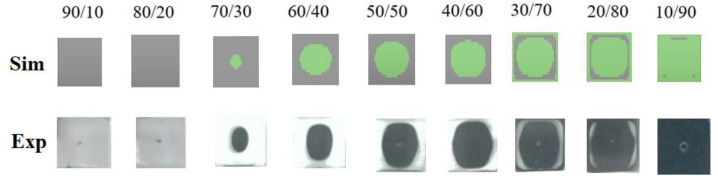
The validation of the cross section of the skin to core with PP/PP arrangement for a skin to core ratio from 90/10 to 10/90.

**Figure 13 polymers-14-04747-f013:**
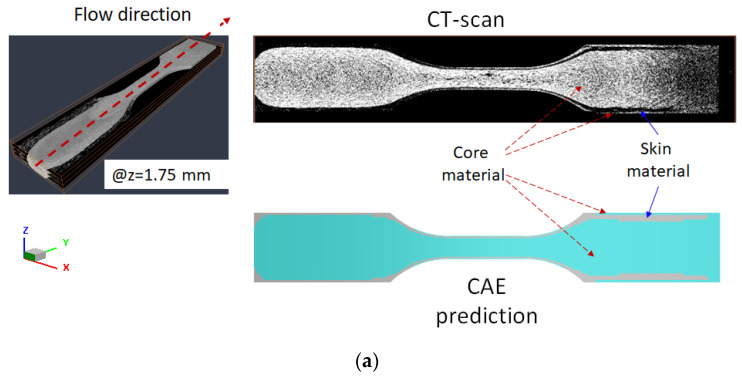
The CAE prediction and experimental examination by micro-CT scan and image analysis on the “core-skin-core” structure for the PP/30SFPP system at the skin to core ratio of 30/70 through cross-section observation. (**a**) XY-plane at central thickness and (**b**) XZ-plane at central region along flow direction.

**Figure 14 polymers-14-04747-f014:**
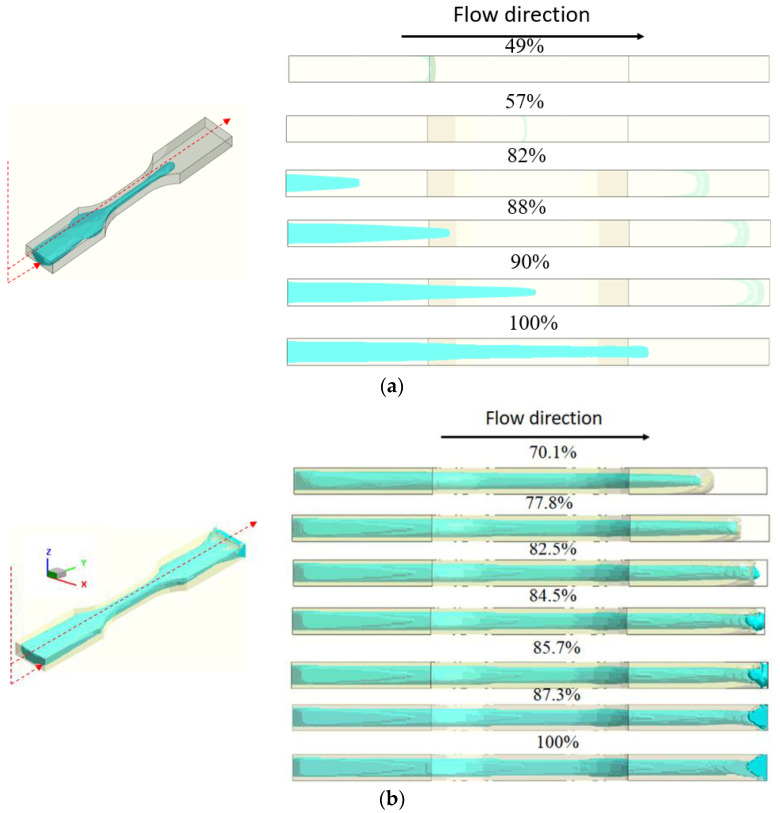
The dynamic behavior for core material advancement: (**a**) at skin to core = 70/30; (**b**) at skin to core = 50/50; and (**c**) at skin to core = 30/70.

**Figure 15 polymers-14-04747-f015:**
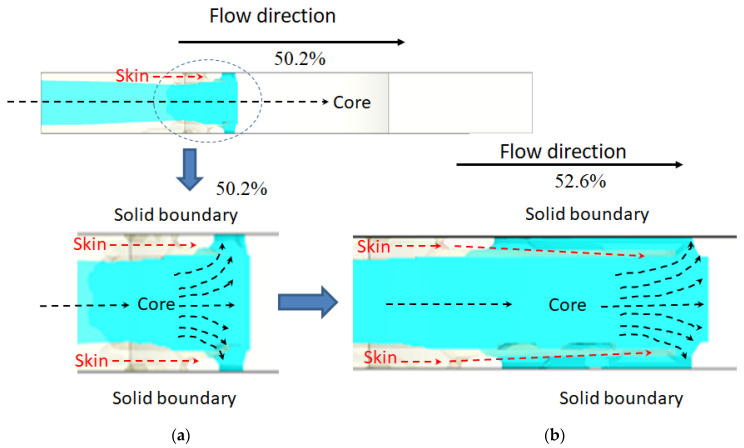
The dynamic behavior for core material advancement at skin to core of 30/70. (**a**) at 50.2% filled and (**b**) at 52.6% filled.

**Table 1 polymers-14-04747-t001:** Materials utilized in this study.

Materials	Content	Grade Name	Producer
PP	PP	Globalene ST868M	LCY Chemical
30SFPP	PP + 30% fiber	Globalene SF7351	LCY Chemical

**Table 2 polymers-14-04747-t002:** Process conditions for basic flow behavior testing.

Filling Time (s)	0.3
Flow rate (cm^3^/s)	10
Melt Temperature (°C)	230
Mold Temperature (°C)	35
Injection Pressure (MPa)	175
Core enters (by volume filled) (%)	60

## Data Availability

The data presented in this study are available on request from the corresponding author.

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
