# Peer review of "Investigation of Parameter Sensitivity and the Physical Mechanism for the Formation of a Core-Skin-Core (CSC) Structure in Two-Stage Co-Injection Molding"

_polymers, 2022, doi:10.3390/polym14214747_

Round 1
Reviewer 1 Report
In this paper, Huang et al. have investigated the formation of the Core-Skin-Core structure and its mechanism in a two-stage co-injection molding process. Numerical simulation and experimental observation have also been applied to carry out this study. The results are interesting. However, the manuscript needs a major revision.
1. The text should be edited carefully. English needs also some review. Some errors are seen here in there and needs to be modified. Care should be given to space between the words.
2. The reference section is too short and needs to cite further papers in this field. There are numerous works performed on the skin-core morphology of the injection molded polymer composites.
3. The following articles are needed to be cited in this article: a) Peculiar crystallization and viscoelastic properties of polylactide/polytetrafluoroethylene composites induced by in-situ formed 3D nanofiber network b) Flow-induced alignment in injection molding of fiber-reinforced polymer composites. c) Correlation between elastic properties and morphology in short fiber composites by X-ray computed micro-tomography. d) Study on mechanical properties of co-injection self-reinforced single polymer composites based on micro-morphology under different molding parameters.
4Authors are highly expected to work further on the reference section.
Author Response
Dear Reviewer,
Thanks for your time and kindness.
Regarding to your priceless suggestions, we have prepared our answers point-by-point. Please see the details in the attached file.
Appreciate very much.
Chao-Tsai Huang

Reviewer 2 Report
The manuscript reports a special polymer-based Core-Skin-Core (CSC) structure, which was discovered in two-stage co-injection molding based on the standard tensile bar (ASTM D638 TYPE V) system. The contents are interesting and fall well within the scope of POLYMERS. Meanwhile, the manuscript is almost well in writing and organization. I recommend its acceptance for publication after minor revisions. My three concerns please be addressed as follows.
The importance of “structure” should be provided sentences as background to support the merits of your job, e.g. multiple-chamber structures, no matter macro-scale or nanoscale, static or dynamic, always play their important roles in the novel materials and their functional performances. Some recent related publications about special structures can be recommended to the readers, e.g. https://doi.org/10.3390/polym14204311, https://doi.org/10.3390/polym13244286, and 10.3390/pharmaceutics14061208.
Several sentences about the scientific and engineering meanings, and the related perspectives can be added in the CONCLUSIONS section for provoking the readers to cite your job after publication.
The references’ formats please be loyal to the journal requests, and to related your job with the most recent publications can benefit a high impact of your article after publications. POLYMERS has a series of related publications.
Author Response
Dear Reviewer,
Thanks for your time and kindness.
Regarding to your wonderful comments, we have prepared our response point-by-point. Please see the details in the attached file.
Appreciate very much.
Chao-Tsai Huang

Round 2
Reviewer 1 Report
N/A